# Is the Stalk of the SARS-CoV-2 Spike Protein Druggable?

**DOI:** 10.3390/v14122789

**Published:** 2022-12-14

**Authors:** Ludovico Pipitò, Christopher A. Reynolds, Giuseppe Deganutti

**Affiliations:** Centre for Sport, Exercise and Life Sciences, Faculty of Health and Life Sciences, Coventry University, Coventry CV1 5FB, UK

**Keywords:** SARS-CoV-2, spike protein, molecular dynamics, fragment-based drug discovery

## Abstract

The spike protein is key to SARS-CoV-2 high infectivity because it facilitates the receptor binding domain (RBD) encounter with ACE2. As targeting subunit S1 has not yet delivered an ACE2-binding inhibitor, we have assessed the druggability of the conserved segment of the spike protein stalk within subunit S2 by means of an integrated computational approach that combines the molecular docking of an optimized library of fragments with high-throughput molecular dynamics simulations. The high propensity of the spike protein to mutate in key regions that are responsible for the recognition of the human angiotensin-converting enzyme 2 (hACE2) or for the recognition of antibodies, has made subunit S1 of the spike protein difficult to target. Despite the inherent flexibility of the stalk region, our results suggest two hidden interhelical binding sites, whose accessibility is only partially hampered by glycan residues.

## 1. Introduction

The spike protein (SP) subunit S1 has been the main drug target in attempts to combat the high infectivity of SARS-CoV-2 by preventing the prime interaction with the human angiotensin-converting enzyme 2 (ACE2) receptor. Since this approach has yet to yield suitable small molecule inhibitors, we have investigated the druggability of the conserved region of the flexible stalk region within subunit S2 by combining molecular docking with high-throughput molecular dynamics simulations. The significance of this work is set against concerns amongst governments, physicians, and the scientific community of the threat that is posed by acute respiratory syndrome coronavirus type 2 (SARS-CoV-2), with almost 530 million cases around the world with more than 6.2 million certified deaths (WHO dashboard, https://covid19.who.int (accessed on 8 June 2022)). SARS-CoV-2 shows a strong affinity for the human angiotensin-converting enzyme 2 (ACE2) receptor, a type 1 transmembrane protein that is responsible for the extracellular conversion of the angiotensin hormone into angiotensin II [1] through the SARS-CoV-2 spike protein (SP). The SP is a highly glycosylated trimeric structure, common amongst the coronaviridae family [2], which is constituted of two subunits, named S1 and S2. While S1 is responsible for molecular recognition of ACE2, S2 is paramount to structural stability and membrane fusion to deliver the viral genome [3]. The efforts of the scientific community were dedicated to promptly developing vaccines or small molecules [4,5,6,7] that are specifically designed to bind and neutralize the area on S1 that is responsible for ACE2 binding, namely the receptor-binding domain (RBD). In 2020, the term variants of concern (VOC) was introduced to describe new SARS-CoV-2 strains which differentiated from the original SARS-CoV-2 wild-type (WT) through mutations that cause drastic enhancement in transmissibility and pathogenicity [8,9,10]. The B.1.617.2 strain (Delta variant) was identified in India by January 2021 and spread rapidly across the globe [11], overcoming the WT. In November 2021, the B.1.1.529 (Omicron variant) became the dominant VOC over the Delta [12].

The SP is constituted of 1273 residues that are divided into two domains. Starting from the signal peptide (residues 1–13) located at the N-terminal, the S1 subunit comprises residues 14–685, while the S2 subunit encompasses residues 686–1273 [13]. Among the SARS-CoV-2 VOCs, major preoccupations regarded those strains that carried important mutations and deletions, especially on the RBD [14]. The vaccine technology that has been developed so far is designed to specifically target the RBD, where the majority of the concerning mutations occur, increasing the risk for antibody inefficacy [15,16,17,18]. The potential loss of efficacy against the Omicron variant was attenuated by a reduced replication and lethality, probably due to Omicron’s inefficiency in exploiting the cellular transmembrane protease serine 2 (TMPRSS2) [19]. However, with the continuous viral diffusion, the likelihood of new mutations remains critical and new variant-specific vaccines need to be developed regularly to keep up with the rate of mutation [20]. While S1 and the RBD are the SP domains that are most prone to mutation, S2 has a higher level of conservation among the coronaviridae family [21]. The only S2 mutations that have been identified so far are N764K, D796Y, N856K, Q954H, N969K, and L981F. Residues I921, S980, V1187, F1218, and I1219 (Figure 1), present in both the Delta and Omicron strains (i.e., the WT bears K921, I980, N1187, L1218, and G1219), are pivotal residues that confer increased flexibility to the stalk [11,22,23]. Interestingly, the region between residues L1145–L1186 (Conserved Region 1, Figure 1), and between E1188–W1217 (Conserved Region 2, Figure 1) contain Loop 1 and Loop2, which contribute to the S1 domain flexibility [24] and exhibit highly-conserved sequences [25].

Molecular dynamics (MD) simulations highlighted crucial aspects and features of the SP [26], including the flexibility of the SP stalk [23], which has been proposed as paramount for ACE2 binding and infectivity [27]. In principle, a small molecule that is able to target the stalk region would be effective on all the VOC by impairing the flexibility of the SP, thus the infectivity of SARS-CoV-2, by interfering with the correct SP-ACE2 alignment. We assessed whether Conserved Region 1 (Figure 1B) is solvent-accessible and able to bind small molecules in the presence of the branched glycans by combining the molecular docking of a library of optimized fragments [28] with high-throughput post-docking MD simulations and binding free-energy calculations. Possible cryptic binding sites were further evaluated through a mixed MD (mixMD) approach [29,30,31]. 

## 2. Materials and Methods

### 2.1. General Workflow

MD simulations and molecular docking were combined in a computational pipeline (Figure 2) aimed to discern potential fragments that are able to bind the SP stalk. Initial MD simulations [32] of the stalk primed the structure for molecular docking, while mixMD [29,30,31] sampled the accessibility of potential pockets. Molecular docking followed by high throughput post-docking MD simulations established the stability of the predicted poses, narrowing the number of putative binders to five, which were evaluated further, in extended 1000 ns MD and mixMD simulations.

### 2.2. Classic MD of the Spike Protein Stalk

The SP stalk was prepared using the CHARMM36 [33,34] force field. The fully glycosylated SP model was retrieved from the CHARMM-GUI repository (https://charmm-gui.org/?doc=archive&lib=covid19 (accessed on 16 January 2022)) [35] and trimmed from residue E1144 to residue W1214, keeping all the glycan residues within this segment. Hydrogen atoms in the S2 domain were added by Propka [36] at a simulated pH of 7.0. The structural integrity was checked through HTMD [37], visually inspected, and disulfide bonds were patched manually through VMD [38] according to previous structural knowledge [3]. Each system was solvated with TIP3P water molecules [39] added to the simulation box considering 15 Å padding in every direction by the Solvate plugin 1.5 (http://www.ks.uiuc.edu/Research/vmd/plugins/solvate/ (accessed on 11 November 2021)). The charge neutrality was achieved by adding Na^+^/Cl^−^ to a concentration of 0.150 M using Autoionize plugin 1.3 (http://www.ks.uiuc.edu/Research/vmd /plugins/autoionize/ (accessed on 11 November 2021)). The initial geometry and potential energy were optimized using the conjugate gradient algorithm by ACEMD3 [40] to eliminate possible clashes and optimize atomic distances. Equilibration was achieved under isothermal-isobaric conditions (NPT) using the Berendsen barostat [41] (target pressure 1 atm) and the Langevin thermostat [42] (target temperature 300 K) with low damping of 1 ps^−1^. During the 4 ns equilibration, a positional restraint of 1 kcal/ mol Å^2^ was applied to the protein alpha carbons for the first 3 ns, and on the protein side chains for the first 2 ns. The production run was performed in the NVT ensemble for 500 ns without any restraint.

### 2.3. Mixed Molecular Dynamics (MixMD)

MixMD [29,31] was used to explore the accessibility of the stalk region to both very low (BENZ, FAC, MTA, Figure 2) and intermediate (fragments 1–3, Table 1) molecular weight molecules, as well as possible cryptic binding sites. The MixMD systems were prepared using PACKMOL [43], setting a minimum distance of 4 Å between each component to avoid clashes and secure a broad placement of the probe molecules. An adequate number of cosolvent molecules were introduced to reach a virtual concentration of 5% *w*/*w*. The systems were then neutralized, equilibrated, and simulated as reported above. For each probe or fragment, the production run was performed in the NVT ensemble for 500 ns. Density maps were computed using the Volmap VMD plugin (https://www.ks.uiuc.edu/Research/vmd/plugins/volmapgui/ (accessed on 11 November 2021)) setting a grid of 0.5 Å while the solvent accessible surface area (SASA) was estimated using vmdICE [44] and Chimera [45].

### 2.4. Fragments Preparation and Molecular Docking

The SpotXplorer database, consisting of 240 optimized molecular fragments [28] (Appendix A), was converted to 3D conformers through the RDKit AllChem module [46], protonated at pH 7 with Chimera, and energy minimized with the RDKit AllChem module. Each fragment was docked using Autodock Vina [47,48], to the Conserved Region 1 of the SP stalk using the structure from the last frame of the cMD equilibration (Appendix A) and the residue V1164 as the center of a grid with a 46 Å side length, for a broad exploration of the stalk surface. For each fragment, ten poses were ranked according to the docking score. Poses that were in contact with glycans or outside the conserved region were discarded. The rationale behind our selection was to narrow down our list of fragments to those that were able to specifically target the conserved set of protein residues of the stalk beyond glycan’s reach. Poses away from the density maps that were produced by BENZ, FAC, and MTA (see below) or not engaging simultaneously with at least two monomers were also excluded.

### 2.5. Post-Docking Classic Molecular Dynamics

The best 500 docking complexes were subjected to 10 ns of post-docking cMD each. The initial CGenFF force field [49,50] topology and parameter files of the molecular fragments were obtained from the CGenFF software. Restrained electrostatic potential (RESP) charges were calculated with AmberTools20 [51] after geometry optimization through Gaussian 09 at the HF/6-31G* level of theory [52]. Each complex was prepared for cMD, equilibrated, and simulated as reported below. For each simulation, similarly to Sabbadin et al. [53], we calculated the dynamic energy score (DES, Equation (1)), which is the sum, over all the MD frames (*n*), of the ratio between the molecular mechanics energy combined with generalized Born surface area (MMGBSA) binding energy and the fragment root mean square deviation (RMSD) to the initial docking pose, using AmberTools20 and VMD.
(1)$$\mathrm{DES}=\sum_{i=1}^{n}\frac{\mathrm{GBSA}}{\mathrm{RMSD}}$$

The use of DES should favor fragments with high binding energy and stability (low RMSD values throughout the MD trajectory). We excluded all the complexes with an average RMSD > 10 Å and RMSD standard deviation > 5 Å and ranked the remaining complexes according to the DES score. We then chose the five best fragments, which were visually inspected to avoid important interactions with glycans. These candidates were further evaluated through 500 ns of cMD (Fragments 1–5, Table 1) or mixMD (Fragments 1–3).

## 3. Results

### 3.1. The Flexible Loops Promote Stalk Flexibility

Computational studies have suggested high flexibility of the SP stalk due to the presence of Loops 1 and 2 [22]. This study focused on the SP stalk Conserved Region 1 (residues L1145 to L1186), which is less glycosylated than other SP domains. During preliminary classic MD (cMD) simulations (Figure 3), the trimeric stalk maintained a stable quaternary structure in the region L1150–L1210, while displaying high flexibility at the level of the N-terminus (residues 1144–1149) and C-terminus (residues 1211–1214). The flexibility at the N and C termini was due to the artificial truncation of S2, adopted to specifically investigate the conserved stalk region. The flexible Loop 1 (residues 1160–1170) was characterized by intermediate flexibility. The high root mean square fluctuation (RMSF, Figure 3A) of the N- and C-termini is ascribable to the artificial cut that was performed to isolate the stalk region from the rest of the SP, which created unnatural protein ends. For this reason, we excluded the four N-termini amino acids for the successive steps of the study. cMD simulations of the stalk in water showed high flexibility and mobility of the trimeric stalk, aided by the opening of transitory pockets within flexible Loops 1 and 2, with the chains temporarily moving away more than 20 Å from each other (Figure 3B) and forming an angle of about 133° between the alpha carbon atoms of P1143–V1164–I1172 (Appendix A). 

### 3.2. The Conserved Region of the Stalk Is Accessible to Solvent and Small Fragments

Preliminary MixMD simulations were employed to assess the accessibility of the Conserved Region 1 (Appendix A). The probes BENZ, FAC, and MTA indicated accessible sections across the stalk. BENZ sampled possible hydrophobic pockets on both the N- (close to K1149) and C-terminal (close to E1195) of the stalk. The latter was explored also by FAC, alongside a further interhelical volume in the proximity of E1182. MTA, and to a lesser extent BENZ, weakly interacted with the stalk at the level of Loop 1 residues V1164–S1170. These results indicate that the glycosylation of the stalk efficiently protects the SP, although some small areas are accessible for potential binding. Interestingly, molecular probes were able to intercalate within the three stalk helices, indicating possible pockets. Solvent accessible surface area (SASA) analysis indicated more solvent-exposed sites at the N-terminal of the stalk, below the connection with S1, and below the flexible Loop 1 (Appendix A) in accordance with mixMD results.

### 3.3. Molecular Docking and High Throughput Post-Docking cMD Predict Interactions with Loop 1

Molecular docking of the optimized library of fragments SpotXplorer that was performed on the fully glycosylated stalk, produced binding poses with scores ranging from −6.7 kcal mol^−1^ (suggested strongly binding) to −2.3 kcal mol^−1^ (suggested weakly binding, Appendix A) with limited convergence with density maps obtained from MixMD simulations of BENZ, FCA, and MTA (Appendix A). We discarded the last half of the poses (poses 1201–2400) as their docking score was higher than the arbitrary value of −4.5 kcal mol^−1^ and all the poses that did not interact with at least two monomers of the stalk or that made contacts with the glycans. The remaining 500 poses were then simulated for 10 ns of MD simulation each, for a total MD sampling of 5 μs, in a fully hydrated and flexible environment. Poses were ranked according to the dynamic energy score (DES, Equation (1), Appendix A). Fragments with DES < −500 kcal mol^−1^ Å^−1^, average RMSD < 10 Å and RMSD standard deviation < 5 Å were retained and visually inspected to filter out fragments that did not remain close to the Conserved Region 1 and made contact with any glycan residue. This led to the retaining of 18 fragments (Appendix A), whose MD simulations were extended.

### 3.4. Longer Post-Docking MD Simulations Rebut Molecular Docking Predictions

Disappointingly, 13 out of the 18 fragments were completely displaced in the first 100 ns of the extended MD simulations (Appendix A). Rotations, openings, and closures of the flexible Loop 1 rapidly disentangled the fragments, causing the rapid displacement of the fragments. However, it is common for molecular fragments to have low affinities [54], sometimes close to or even lower than binding assay sensitivity [55]. The remaining five compounds (Fragments 1–5, Figure 4, Table 1) initially interacted with Loop 1, in correspondence with residues N1158–S1170, before moving away from the initial position in less than 300 ns (Figure 4). Planar compounds, predicted by molecular docking in sites that were also sampled by the mixMD of BENZ and FAC, displayed better interactions with Loop 1 residues N1158–S1170. Fragments 1–3 resided at the center of the trimer for more than 100 ns, before dissociating; Fragments 4 and 5 resided mainly on the C-terminal end of Loop 1 before unbinding through a temporary tunnel formed between the stalk chains.

### 3.5. MixMD to Check the Fragments’ Accessibility to Conserved Region 1

Since mixMD simulations of the molecular probes BENZ, FAC, and MTA (Appendix A) suggested some degree of accessibility to the stalk despite the high glycosylation of the SP, we ran further mixMD simulations using Compounds 1–3 (Table 1, Figure 4) to investigate the accessibility of larger compounds, and any convergence with the metastable configurations that were sampled during the 500 ns post-docking cMD simulations. During mixMD simulations, the fragments were able to reach the stalk protein surface on isolated spots, overcoming the shield that was provided by the glycans (Appendix A). Fragment 1 made fewer interactions with the stalk among the three compounds (Appendix A), mainly engaging residues F1148, T1155, K1149, E1151, and L1152 (Appendix A). Fragment 2 formed interatomic contacts with residues K1149, Y1155, E1151, L1152, V1176, H1159, and F1148 (Appendix A, Appendix A). Compound 3 formed the most persistent interactions with the stalk, engaging Y1155, E1182, E1151, N1178, Q1180, K1149, H1159, L1152, R1185, ASP1153, F1156, V1177, and D1153 side chains (Appendix A, Appendix A). Many of these residues were engaged with a high turnover by different fragment molecules, indicating low stability of the interactions. However, mixMD simulations indicated two cryptic binding sites; these sites were occupied for almost the whole duration during single molecule simulations of 1–3. In Sub-pocket 1 (Figure 5B), located in the Conserved Region 1, Fragment 3 formed hydrogen bonds with E1151 and hydrophobic contacts with residues F1148, K1149, L1152, and Y1155. In Sub-pocket 2 (Figure 5C, Videos S1 and S2) at the interface between Conserved Regions 1 and 2, a molecule of three formed hydrophobic contacts with L1186, V1189, A1190, and K1191, and 3 hydrogen bonds with E1182, N1194, and N1187; glycan residues participated in the stabilization of the molecule through van der Waals contacts (Video S1). Sub-pocket 2 was also identified by the FAC probes around residues E1182–K1191 (Appendix A), suggesting that it might provide an overall stable binding site, although far from the flexible Loop 1. MixMD simulations of 3 sampled an alternative conformation of the fragment within Sub-pocket 2; interestingly, three molecules of Fragment 3 occupied putative binding sites on the symmetric trimeric stalk at the same time (Video S2), suggesting accessibility and the possibility to exploit the two pockets for drug design purposes.

## 4. Discussion

The SP stalk region is conserved amongst the SARS-CoV-2 VOCs. Given its role in orienting the RBD for binding to ACE2, impairing its flexibility through the binding of a small molecule or disrupting its integrity through a planar molecule that is able to intercalate between the stalk helices could represent a therapeutic approach to explore. We investigated the druggability of the SP stalk using 240 biologically active molecular fragments, evaluating the shielding effect of the glycan residues on the protein surface. Our computational workflow combined molecular docking, high-throughput cMD simulations, and mixMD as orthogonal methods to evaluate putative interactions on the stalk region. Molecular docking predicted putative interaction sites around residues T1160–S1170. High-throughput cMD simulations of 500 docking poses suggested the instability of docking predictions, except for a few fragments that were then further evaluated in longer simulations. Metastable interactions on the whole stalk were confirmed in the proximity of residues H1159–I1169. The mixMD simulations of the three most promising fragments 1–3 sampled two narrow binding sites within the helices of the stalk. Sub-pocket 1 (Figure 5B) is located close to Loop 1 and might represent an anchor point for the design of larger ligands bearing a group that intercalates between the stalk helices and a flexible moiety that impairs Loop 1 dynamics and likely the whole SP flexibility, although this strategy appears complicated due to the extreme flexibility of the unstructured region as well as the possible influence of the glycans on the S1 unit. Sub-pocket 2 (Figure 5C) presents a balanced mix of hydrophilic and hydrophobic residues that increase the druggability compared to Sub-pocket 1. However, it bears the non-conserved residue N1147, it is more distant from Loop 1 than Pocket 1, and it is heavily influenced by glycan residues.

## 5. Conclusions

We applied a computational pipeline that exploited hundreds of classic MD simulations to assess the best molecular docking poses according to the Vina scoring function. Despite the good scores that were obtained, most of the poses displayed instability in a fully flexible and hydrated simulated environment. This should stress the importance of combining MD simulations with molecular docking for drug discovery and development. Despite the state-of-the-art methodology that was used, our work highlights the challenges of exploiting the SP stalk as a therapeutic target. A fragment-based approach appears challenging for this task whereas alternative routes that consider the symmetry of the stalk and its sub-pockets might improve the results that were obtained by adding up binding stabilization from more interaction sites. Future studies could, therefore, be based on the symmetry of the trimeric SP stalk and consider higher molecular weight ligands, to evaluate if they are able to overcome the glycan residues’ hindrance of the binding pathway to the pockets that were identified as Sub-pocket 1 and Sub-pocket 2. This work, to the best of our knowledge, is the first computational characterization of the SP stalk from a drug discovery perspective.

## Figures and Tables

**Figure 1 viruses-14-02789-f001:**
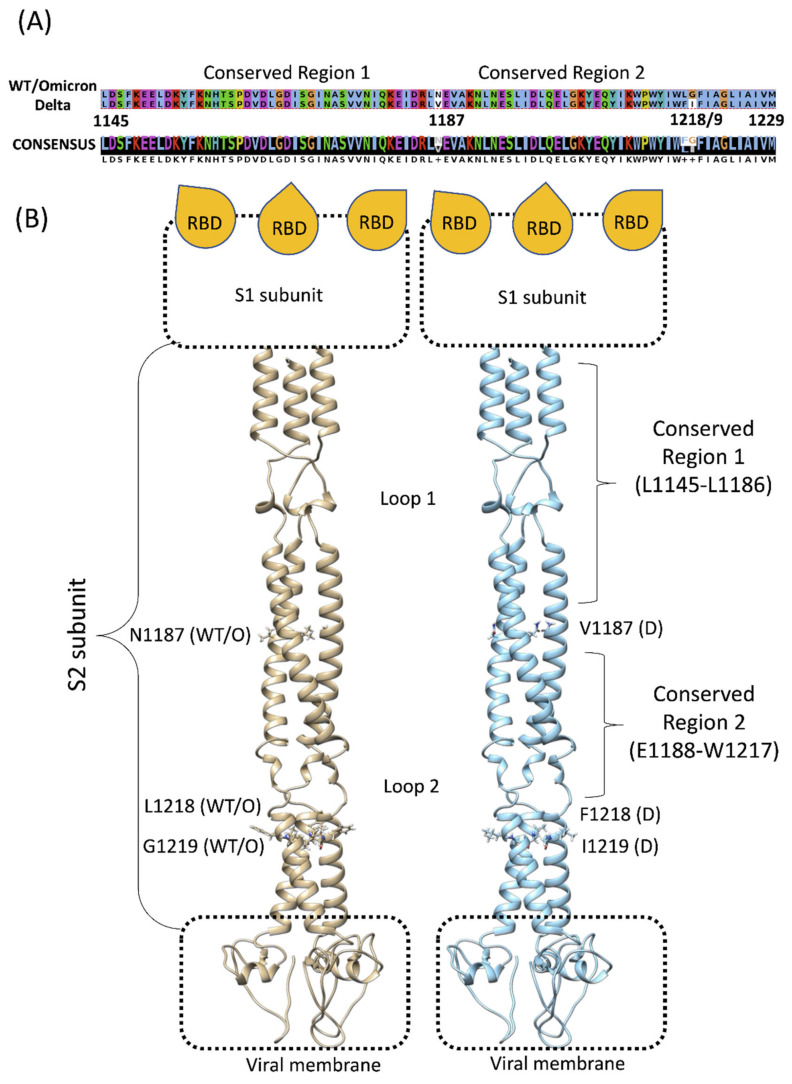
(**A**) Sequence alignment between the conserved SP stalk region of the wild-type (WT) or Omicron and Delta variants. (**B**) Structural comparison between WT or Omicron (tan ribbon) and Delta S2 (cyan ribbon) regions. The segment between residues L1145–W1217 is almost identical between the strains, except for residue 1187, which divides the Conserved Regions 1 and 2. Glycans were removed for clarity; the S1 subunit with the three receptor-binding domains (RBDs) and the viral membrane are schematically represented.

**Figure 2 viruses-14-02789-f002:**
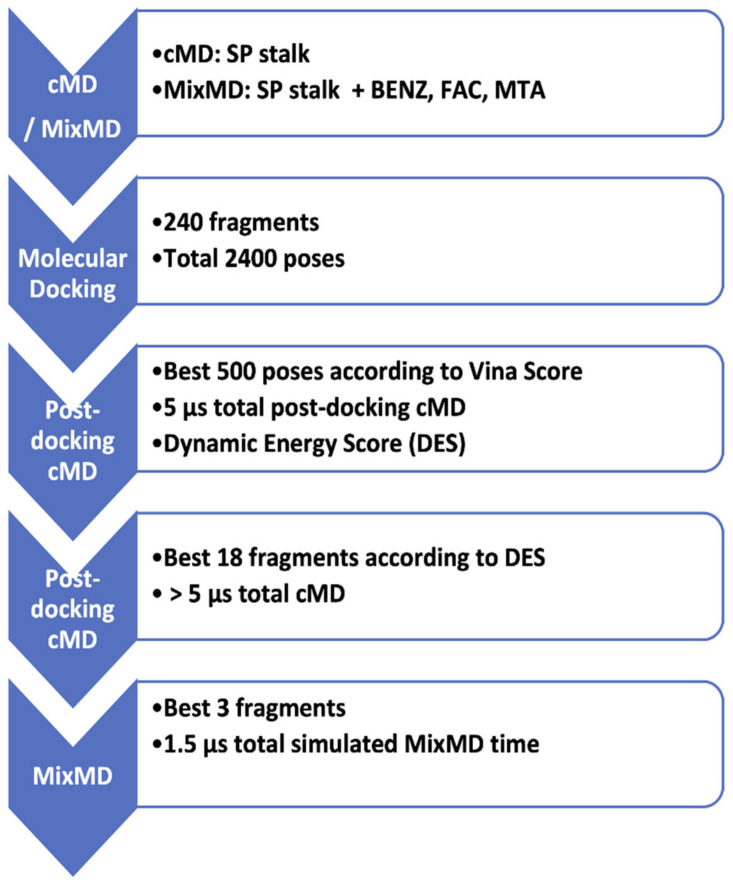
Computational workflow of the study. Preliminary classic molecular dynamics (cMD) and mixed MD (mixMD) were performed on the spike protein stalk (SP); an MD-extracted conformation of the SP stalk was used to dock 240 molecular fragments. The best 560 poses according to Vina score were subjected to 10 ns of cMD ed evaluated according to the dynamic energy score (DES, Equation (1)). The best five fragments according to DES were further simulated through cMD and mixMD. BENZ: benzene; FAC: formic acid; MTA: methylamine. These fragments contain the basic functional groups, namely aromatic (benzene, BENZ), hydrogen bond donors (formic acid, FAC; methylamine, MTA), hydrogen bond acceptors (FAC), and aliphatic hydrophobic (MTA).

**Figure 3 viruses-14-02789-f003:**
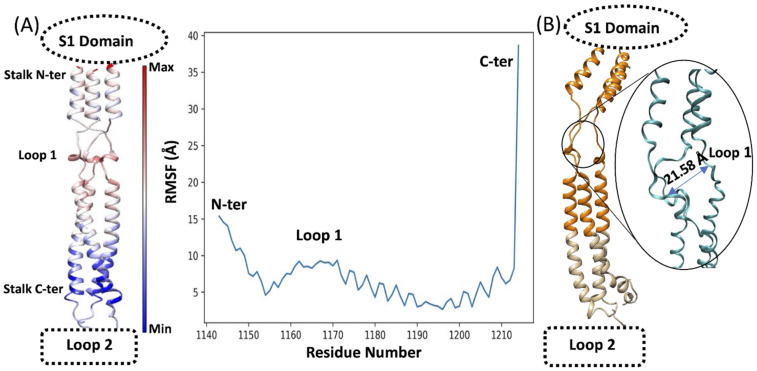
(**A**) Root mean square fluctuation (RMSF) of the SP stalk during cMD simulations. The RMSF of each residue is mapped on the structure (left panel) and color-coded according to the value (flexible residues are red); RMSFs are also plotted on the sequence (right panel); (**B**) the SP stalk Conserved Region 1 (orange) is subjected to high flexibility during cMD simulations at the level of Loop 1; the position of the S1 domain and the Loop 2 is reported for reference.

**Figure 4 viruses-14-02789-f004:**
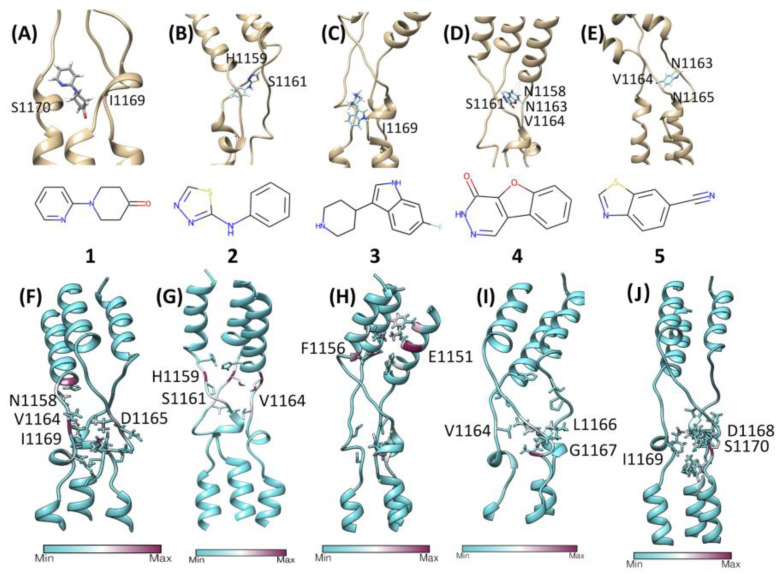
(**A**–**E**) The best five fragments (1–5) according to molecular docking followed by 10 ns of post-docking cMD simulations; the final conformation (stick representation) after 10 ns is reported. (**F**–**J**) Stalk residues (ribbon) that most interacted with Fragments 1–5 during 500 ns of cMD; residues with the highest occupancy are in maroon.

**Figure 5 viruses-14-02789-f005:**
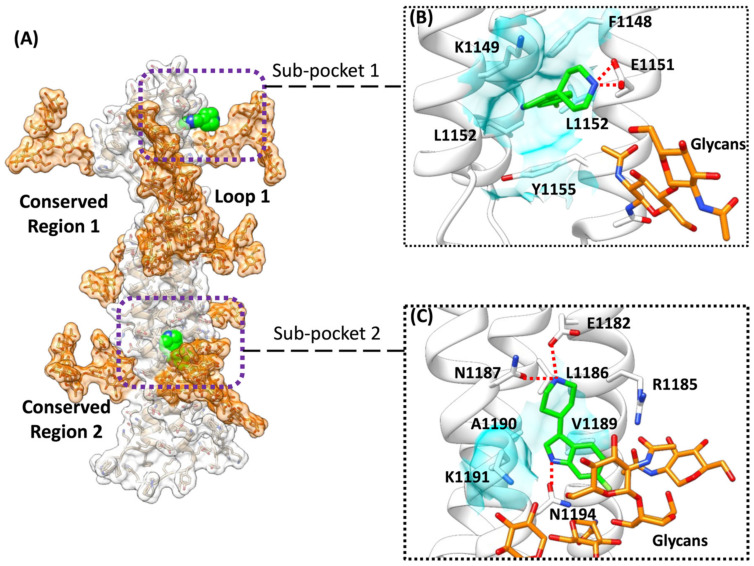
MixMD simulations indicated two sub-binding pockets. (**A**) Overview of the stalk SP (white transparent surface); glycan residues are depicted with an orange transparent surface, while Fragment 3 is in green van der Waals spheres. (**B**) Magnification of the Sub-pocket 1 with 3 in green stick representation; (**C**) magnification of the Sub-pocket 2; Fragment 3 is in green stick representation, while glycan residues are orange. Hydrogen bonds are shown as dashed red lines.

**Table 1 viruses-14-02789-t001:** Summary of the best five fragments after 500 ns of cMD.

Fragment	IUPAC NAME	Pre-Unbinding Position	Displacement Time
1	1-pyridin-2-ylpiperidin-4-one	N-terminal	~160 ns
2	*N*-phenyl-1,3,4-thiadiazol-2-amine	N-terminal	~200 ns
3	6-fluoro-3-piperidin-4-yl-1H-indole	N-terminal	~300 ns
4	3H-[1]benzofuro[2,3-d]pyridazin-4-one	C-terminal	~200 ns
5	1,3-benzothiazole-6-carbonitrile	C-terminal	~110 ns

## Data Availability

Raw data were generated in this study are available from the corresponding author GD on request.

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
