# Peer review of "Is the Stalk of the SARS-CoV-2 Spike Protein Druggable?"

_viruses, 2022, doi:10.3390/v14122789_

Round 1

Reviewer 1 Report

The authors performed a high-quality molecular dynamic simulations beyond docking identifying possible binding sites. The results are interesting and the paper is well written. I think this paper meets the high standards of Viruses. Few minor points should be addressed by the authors before publication.

Minors:

Typos, spaces or repetitions along the manuscript need to be revised

i.e.

Line 44: rapidly […] rapidly

Line 58: “conserved in both Delta and Omicron”. Did you mean WT (instead of Delta)?

Line 170: “terminal four N-terminal”

etc

In Figure 1 add “S2 subunit” to make it easier to visually locate where the subunit is

It would be helpful in the Introduction to define the amino-acids length comprised in S1 and S2.

IN M&M please specify the length of the simulation in all the Molecular Dynamics sections.

In the section “Fragment preparation and molecular docking” how where fragments optimized?

Author Response

The authors performed a high-quality molecular dynamic simulations beyond docking identifying possible binding sites. The results are interesting and the paper is well written. I think this paper meets the high standards of Viruses. Few minor points should be addressed by the authors before publication.

We thank the Reviewer for the positive comment on our manuscript.

Minors:

1) Typos, spaces or repetitions along the manuscript need to be revised

i.e.

Line 44: rapidly […] rapidly

The suggested repetition was amended.

Line 58: “conserved in both Delta and Omicron”. Did you mean WT (instead of Delta)?

We thank the Reviewer for pointing this out. Residues I921, S980, V1187, F1218, and I1219 belong to both the Delta and Omicron variants. The WT has K921, I980, N1187, L1218, and G1219 instead. This is now explicitly reported in the text.

Line 170: “terminal four N-terminal”

etc

We have thoroughly checked the manuscript as suggested.

2) In Figure 1 add “S2 subunit” to make it easier to visually locate where the subunit is

We have explicitly included the S2 subunit reference to the figure for a more complete overview of the structure.

3) It would be helpful in the Introduction to define the amino-acids length comprised in S1 and S2.

We have now introduced a general description of the two SP domains and added the appropriate reference as requested.

4) IN M&M please specify the length of the simulation in all the Molecular Dynamics sections.

We specified the length of MD simulations in all the Methods sections concerned.

5) In the section “Fragment preparation and molecular docking” how where fragments optimized?

We have specified what RDkit module was used to optimize the 3D geometry of the fragments (namely the AllChem module).

Reviewer 2 Report

I recommend accepting this paper with minor revisions for publication at this esteemed Journal. I just have a minor issue to clarify  these  findings. 

As of line 164 "
the trimeric stalk maintained a stable 164 quaternary structure in the region L1154-L1166
In this line it was mentioned that this region is stable and conserved region. But in the next line "
The 166 flexible Loop 1 (residues 1160-1170) was characterized by intermediate flexibility". Please explain it . 
As a suggestion some places author should add figure and tables itself in the manuscript unlike keeping it as a supplementary data, should be add in the main manuscript. Like Figure S2 page 6 ., Table S4 page 7. Also if author can improve the conclusion section. 

I am not recommending any additional experiments.

Thanks 

Author Response

I recommend accepting this paper with minor revisions for publication at this esteemed Journal. I just have a minor issue to clarify  these  findings
We thank the Reviewer for the positive comment on our manuscript.

1) As of line 164 " the trimeric stalk maintained a stable 164 quaternary structure in the region L1154-L1166" 
In this line it was mentioned that this region is stable and conserved region. But in the next line "The 166 flexible Loop 1 (residues 1160-1170) was characterized by intermediate flexibility". Please explain it .

We thank the Reviewer, this text passage could be improved indeed. We have amended the text to clarify the point.

2) As a suggestion some places author should add figure and tables itself in the manuscript unlike keeping it as a supplementary data, should be add in the main manuscript. Like Figure S2 page 6 ., Table S4 page 7.

We share the Reviewer’s point as we carefully evaluated if incorporating some supplementary information in the main text, before the original submission. However, we decided to keep the number of Figures to a maximum of 5. At this stage, we respectfully prefer not to alter the structure of the manuscript.

3) Also if author can improve the conclusion section. 

Some further considerations have been added to the conclusion as suggested by the Reviewer.